# Self-Injuries and Their Functions with Respect to Suicide Risk in Adolescents with Conduct Disorder: Findings from a Path Analysis

**DOI:** 10.3390/jcm10194602

**Published:** 2021-10-07

**Authors:** Monika Szewczuk-Bogusławska, Małgorzata Kaczmarek-Fojtar, Joanna Halicka-Masłowska, Błażej Misiak

**Affiliations:** 1Department of Psychiatry, Wroclaw Medical University, 50-367 Wrocław, Poland; joanna.halicka-maslowska@student.umed.wroc.pl (J.H.-M.); blazej.misiak@umed.wroc.pl (B.M.); 2Department of Emergency Medicine, Wroclaw Medical University, 50-556 Wrocław, Poland; gosia.kaczmarek@gmail.com

**Keywords:** self-injuries, suicidality, adolescence, mental disorders

## Abstract

Non-suicidal self-injuries (NSSIs) have been identified as one of the most predictive factors of suicidal behaviours in adolescents. However, it remains unknown whether certain functions of NSSIs are associated with suicide risk, and what are the underlying mechanisms. Therefore, we aimed to investigate the association between functions of NSSIs and suicide risk in adolescents with conduct disorder (CD), which shares some common characteristics with NSSIs. Participants were 215 adolescents (155 females, 72.1%) with CD. Functions of NSSIs, depressive symptoms, the levels of impulsivity, anxiety, self-esteem and aggression were examined. There were 77 adolescents with lifetime history of NSSIs (35.8%). Among them, adolescents with lifetime history of suicide attempt were significantly more likely to report anti-dissociation and anti-suicide function of NSSIs. They had significantly higher levels of anxiety as well as significantly lower self-esteem. Higher lifetime number of NSSIs was associated with higher odds of reporting anti-dissociation and anti-suicide functions. Moreover, these two functions fully mediated the association between lifetime number of NSSIs and suicide risk after co-varying for depressive and anxiety symptoms as well as self-esteem. The present findings indicate that anti-suicide and anti-dissociation functions of NSSIs might be crucial predictors of suicide risk in adolescents with CD.

## 1. Introduction

Suicidality among adolescents remains one of crucial concerns in both medical and social fields of modern societies. As a result, intensive research is ongoing to establish factors that predict suicidal behaviours in young people. Non-suicidal self-injuries (NSSIs) have been identified as one of the strongest risk factors of suicidal behaviours in adolescents according to previous studies [1,2]. Younger age at onset of NSSIs, in comparison with suicide attempts, has been replicated by studies conducted in the general population and in patients diagnosed with mental disorders, indicating that NSSIs precede suicidal behaviours [3,4,5]. Taking into account significantly higher prevalence of NSSIs in comparison with suicidal attempts estimated in the general population of adolescents (17.0–18.0% vs. 4.1–6.7%), different characteristics of NSSIs and other variables that might potentially affect the link between NSSIs and suicide attempts are extensively being investigated [6,7].

High frequency of NSSIs, higher number of methods approached to self-injure, cutting, duration of NSSIs (longer than one year), severe damage of tissues, co-occurrence of suicidal intention, self-disclosure of suicidal thoughts as well as the absence of pain have been associated with a risk of suicidal behaviours [8,9,10,11]. Furthermore, functions of NSSIs seem to play an important role in shaping suicidal behaviours. Functions of NSSIs are defined as a motivational component of an act. People who engage in NSSIs are motivated by numerous inter- or intrapersonal reasons. Intrapersonal functions are more prevalent and serve mainly as an emotion regulating strategy [12]. Intrapersonal functions are represented, among others, by the anti-suicide function, which has been suggested to affect the association between NSSIs and suicide attempts. However, two contraindicatory approaches to the relationship between anti-suicide function of NSSIs and suicide attempts are postulated. On the one hand, according to Suyemoto [13], the protective effect of anti-suicide function is considered. On the other hand, the presence of the anti-suicide function indicates concomitant suicidal ideation, which, per se, is related to suicidality. In other words, engagement in NSSIs, in order to resist from suicide attempt, might be understood as a risk factor of suicide due to co-occurrence of psychopathology (suicidal thoughts), which might also be a symptom of concomitant mental disorders (e.g., depression or borderline personality disorder). The results of empirical research are also inconclusive. Some authors suggest the protective role of anti-suicide function while others indicate that this function might predict higher risk of suicide attempts. For instance, Burke et al. identified the anti-suicide function as one of the most important predictive features of suicide attempts in a large suicide attempt sample of young adults with a history of NSSIs [14]. In turn, Robinson et al. showed that three functions of NSSIs (self-punishment, anti-suicide and sensation-seeking functions) are associated with more severe suicidal behaviours in adolescents from the community sample [15]. Paul et al. showed the strongest association between anti-suicide function and suicide attempts in the large sample of university students and indicated another intrapersonal function of NSSIs, anti-dissociative feeling generation function, as potentially crucial in the association between NSSIs and suicidal behaviours. The authors found that regardless of the fact that the most common functions of NSSIs in different suicidal groups (ideation, plans and attempts) were reasons related to reducing negative emotions, the relationship between anti-dissociative function and suicide attempts was stronger than the effect of the most common functions [16]. However, the majority of previous studies have been based on heterogenous criteria of NSSIs and suicidal behaviours. Thus, most recent studies aim to analyse the association between new diagnostic categories developed in the DSM-5 for further studies: non-suicidal self-injury disorder (NSSID) and suicidal behaviour disorder (SBD)-defined as at least one suicide attempt during past 24 months [17]. High co-occurrence of NSSID and SBD was found in a suicide sample of psychiatric inpatients, where even 51.2% patients with NSSID might meet the SBD criteria, and 65.6% individuals diagnosed with SBD meet the NSSID criteria [15]. Our recent study of adolescent girls diagnosed with conduct disorder (CD) estimated the prevalence of SBD at 50.0% in individuals with NSSID, and NSSID was present in 52.2% patients with SBD [5].

It is worth noting that the anti-suicide function of NSSIs is not listed as a separate function of NSSID in the DSM-5 criteria. Thus, in the recent study of adolescent, female inpatients, Kraus et al. added to existing NSSID criteria the anti-suicide function and found that one third of the individuals diagnosed with NSSID endorse NSSIs as a method of avoiding suicide. However, the comparison of individuals reporting anti-suicide function and those who had not declared this function did not reveal significant between-group differences in the association of NSSIs with suicidal ideation and attempts. Interestingly, the diagnosis of SBD according to DSM-5 was more prevalent in subjects reporting the anti-suicide function (72.22% vs. 55.26%). Nevertheless, these results did not support the concept that the anti-suicide function of NSSIs specifically affects the association between NSSID and suicidal attempts [18].

The discrepancies in the results of the effect of anti-suicide function of NSSIs on suicidal behaviours may result from several reasons. First, different definitions or criteria are used to establish the diagnosis of NSSIs (NSSIs as the phenomenon beyond current diagnostic systems vs. NSSID as a disorder). Second, various tools are used to assess characteristics of NSSIs. Third, different populations are examined (community vs. clinical suicide samples, adolescents vs. adults). From a clinical point of view, identifying patients who might commit suicide is crucial to effectively prevent this outcome of mental disorders. Due to the varied psychopathology of mental disorders, specific disorders might differ regarding the association between characteristics of NSSIs and suicidal behaviours. Therefore, there is a rationale to analyse the association between NSSIs and suicidal behaviours in a homogenous sample, in terms of psychiatric diagnosis. Conduct disorder (CD) shares some common characteristics with NSSIs including the following: aggression (which has been suggested as one of the crucial risk factors of NSSIs), a typical age of onset in adolescence as well as a history of interpersonal difficulties, childhood maltreatment, an invalidating family environment and low socioeconomic status [19,20,21,22].

According to our previous study of female adolescents diagnosed with CD, which revealed high prevalence of SBD and showed that a diagnosis of NSSID and depressive symptoms are associated with SBD, we decided to extend our research in this clinical group. Building upon existing evidence from our studies, we decided to extend our sample and include both females and males with CD employing various functions of NSSIs. In the present study, we aimed to explore what functions of NSSIs are associated with a risk of suicide in adolescents with CD. Moreover, we tested the hypothesis that certain functions of NSSIs might mediate the association between their severity in terms of frequency and a risk of suicide.

## 2. Materials and Methods

### 2.1. Participants

This study is a continuation of our previous project on a female sample, whose results were published elsewhere [5]. The studied group was extended to enlarge the sample and to include male participants. Finally, the studied group consists of individuals who completed all measures (71 females who participated in the first study and 144 individuals who participated in the extension study). A total of 215 adolescents (155 females and 60 males) diagnosed with CD were approached for participation (15.1 ± 1.2 years). All participants were residents of the Youth Sociotherapy Center (YSTC) No. 2 in Wroclaw (Poland). YSTCs are facilities established by the Polish Ministry of Education to support children and adolescents who present developmental or educational problems or are diagnosed with CD. YSTCs provide round-the-clock care of professional staff, schooling, accommodation, psychotherapy and sociotherapy.

### 2.2. Procedures

The study procedures were approved by the Bioethics Committee of at Wroclaw Medical University (Reg. No KB-352/2013 and 532/2017). All individuals who participated in the study and their parents or legal caregivers gave a written informed consent for participation. The study was performed according to the Declaration of Helsinki.

All participants signed a paper version of informed consent during the first in-person meeting with a psychologist or child and adolescent psychiatrist, who explained all study procedures and discussed them with the patient. The consent was prepared according to the requirements of the Bioethics Committee at Wroclaw Medical University (Poland). In the second part of the meeting, each participant was examined by a trained investigator (psychologist or child and adolescent psychiatrist) to obtain all data required to establish a diagnosis of CD. Next, as a part of a diagnostic process, the Polish version of the Mini International Neuropsychiatry Interview for Children and Adolescent (MINI-KID) was administered.

During the next meeting, data regarding NSSIs were obtained using a semi-structured interview (see Table A1 in Appendix A for details). Furthermore, all participants filled in a set of questionnaires to assess the level of impulsivity, depressive symptoms, anxiety, aggression and self-esteem. To collect data regarding psychopathology, the following measures were used: (1) The Eysenck’s Impulsivity Inventory (IVE), (2) the Children’s Depression Inventory (CDI-2), (3) the Spielberg State-Trait Anxiety Inventory (STAI), (4) the Buss–Perry Aggression Questionnaire (BPAQ) and (5) the Rosenberg Self-Esteem Scale (SES).

### 2.3. Measures

#### 2.3.1. The MINI-KID 

The MINI-KID [23] is a structured diagnostic interview designed to assess the symptoms of mental disorders according to the DSM-IV and ICD-10 criteria in children and adolescents aged 6 to 17 years. The MINI-KID can be administered by interviewing exclusively adolescent respondents or by interviewing both the child/adolescent and the parent(s) or caregiver(s). Good psychometric properties of the MINI-KID were shown in the study of the Polish version with comparable validity parameters to those obtained by other studies [24].

#### 2.3.2. NSSIs-Interview

The semi-structured interview was used as a part of a psychiatric examination to collect data on characteristics of NSSIs (age of onset, number, functions, methods and purposes of NSSIs). The interview questionnaire is shown in the Table A1.

Self-injuries were classified as NSSIs if they met the following characteristics: 1. Deliberate self-injury of body tissues without suicidal intent during the act; 2. Self-injuries with at least one of the methods: cutting, burning, scratching, banging, excessive rubbing and hitting; 3. At least one of the following functions of self-injuries was indicated by the patient: relieving an interpersonal difficulty, reducing negative emotions and/or inducing positive feelings.

Therefore, the following answers were required in the semi-structured interview about NSSIs:-“Yes” to question 1 and “No” to question number 2.-At least one of the following: cutting, burning, picking, hitting and/or excessive rubbing indicated as the answer to question number 3.-One of the following answers to question number 5 (in subsection statements): to alleviate anxiety/sadness/tension, to manage bad feelings, to punish or control somebody, to get someone’s attention, to punish yourself and/or to feel better.

#### 2.3.3. The Inventory of Statements about Self-Injury (ISAS)

The second section of the ISAS was used to collect data regarding functions of NSSIs. The ISAS consists of 39 statements that assess the following functions: (1) affect-regulation, (2) anti-dissociation feeling generation, (3) anti-suicide, (4) autonomy, (5) interpersonal boundaries, (6) interpersonal influence, (7) marking distress, (8) peer-bonding, (9) self-care, (10) self-punishment, (11) revenge, (12) sensation seeking and (13) toughness. Three statements are assigned to each function and rated as follows: 0—not relevant, 1—somewhat relevant and 2—very relevant [25]. Psychometric parameters of the Polish version of ISAS were estimated as good by Kubiak [26].

#### 2.3.4. The Eysenck’s Impulsivity Inventory (IVE)

The IVE consists of 63 diagnostic questions. The answers are based on yes/no responses. The tool includes three subscales: impulsivity, venturesomeness and empathy [27]. The Cronbach’s alpha for each subscale was as follows: 0.75 (impulsivity), 0.66 (venturesomeness) and 0.65 (empathy).

#### 2.3.5. The Children Depression Scale (CDI-2:S)

This questionnaire consists of 28 items. The CDI-2:S version is used to perform a comprehensive assessment of depressive symptomatology as well as emotional and functioning problems in children and adolescents. The tool includes self-rating subscales measuring emotional problems and problems in daily functioning, interpersonal problems, a lack of behaviour efficacy, negative mood, somatic symptoms and low self-esteem [28]. The Cronbach’s alpha for CDI-2:S was 0.94 in our sample.

#### 2.3.6. The Spielberg State-Trait Anxiety Inventory (STAI)

The STAI contains two subscales: state (X1) and trait anxiety (X2). State anxiety (X1) is described as the emotional status, which varies across time, and depends on circumstances. In turn, the level of trait anxiety (X2) is stable and is associated with experiencing emotions as a feature of personality. The scale consists of 40 items that are rated on a Likert-like scale scored between 1 and 4 [29]. In our sample, the Cronbach’s alpha was 0.94 for state anxiety and 0.99 for trait anxiety.

#### 2.3.7. The Rosenberg Self-Esteem Scale (SES)

The SES is a self-rated questionnaire for global self-esteem assessment. It measures positive and negative feelings about the self. The questionnaire consists of 10 statements that are based on a 4-point Likert scale. The person completing the scale may strongly agree, agree, disagree or strongly disagree with 10 statements regarding their self-worth [30]. The Cronbach’s alpha for the SES total score was 0.89.

#### 2.3.8. The Buss-Perry Aggression Questionnaire (BPAQ)

The BPAQ is a self-report rating questionnaire that consists of 29 items and is scored on a 5-point Likert-like scale (1- “extremely uncharacteristic of me” to 5- “extremely characteristic of me”). A range of aggressive symptoms are measured by specific subscales (physical aggression, verbal aggression, anger and hostility) [31]. The Cronbach’s alpha for each BPAQ subscale was as follows: 0.80 (for physical aggression), 0.73 (for verbal aggression), 0.62 (for anger) and 0.77 (for hostility).

### 2.4. Data Analysis

First, in case of continuous variables, data distribution was analysed using the Kolmogorov–Smirnov test. Next, in case of non-normal distribution, bivariate comparisons were tested using the Mann–Whitney U test. Otherwise, the t-tests were employed. Between-group differences with respect to categorical variables were analysed using the chi-square test. The Spearman rank correlation coefficients were explored to test associations between continuous variables. The level of significance in case of bivariate tests was set at *p* < 0.05. The PROCESS Macro Model 4 was employed to perform mediation analysis [32]. Separate models for distinct mediators were run to avoid potential multicollinearity (Figure 1). Lifetime number of NSSIs was included as an independent variable while a history of suicide attempts was an outcome variable. According to assumptions of mediation analysis, the mediator must be associated with the independent variable and the outcome variable. Therefore, potential mediators were selected from the functions of NSSIs that were significantly associated with lifetime number of NSSIs and lifetime history of suicide attempts. Age, sex and the levels of anxiety and depressive symptoms were added as co-variates. The bootstrap calculation with 5000 samples was performed to check direct and indirect effects. Mediation was considered significant if the 95% CI of indirect effect did not include zero. All analyses were carried out using the Statistical Package for Social Sciences, version 20 (SPSS Inc., Chicago, IL, USA).

## 3. Results

General characteristics of the sample with respect to a lifetime history of suicide attempts are presented in Table 1. Seventy-seven adolescents (35.8%) confirmed engaging in NSSI during their lifetime. Compared to individuals without lifetime history of NSSIs, these adolescents were significantly younger (14.6 ± 1.1 vs. 15.1 ± 1.3 years, *U* = 1830.5, *p* = 0.035) and were significantly more likely to be females (74.0% vs. 35.6%, *χ*^2^ = 19.7, *p* < 0.001). Adolescents with lifetime history of suicide attempts had a significantly higher recent-year number of NSSIs and levels of anxiety. In turn, the level of self-esteem was significantly lower in this subgroup of participants. Adolescents with a lifetime history of suicide attempts were significantly more likely to employ anti-suicide and anti-dissociation functions of NSSIs. There were significant positive correlations between lifetime number of NSSIs and the odds of employing the following functions of NSSIs: affect regulation, self-punishment, self-care, anti-dissociation feeling generation, anti-suicide and toughness (Table 2).

Results of mediation analyses are shown in Table 3. There was a significant effect of lifetime number of NSSIs on anti-dissociation feeling generation. The effect of this function on lifetime history of suicide attempts and direct effect of lifetime number of NSSIs on lifetime history of suicide attempts were not significant. However, the indirect effect (through anti-dissociation function) was significant, indicating full mediation. Similar results were obtained with respect to the anti-suicide function of NSSIs. Indeed, the direct effect of lifetime number of NSSIs on anti-suicide function and the effect of anti-suicide function on lifetime history of suicide attempts were significant. The direct effect of lifetime number of NSSIs on lifetime history of suicide attempts was not significant. In turn, the indirect effect (through anti-suicide function) was significant, indicating full mediation. Similar findings were obtained after excluding participants with lifetime number of NSSIs higher than 100 (sensitivity analysis, Table 4).

## 4. Discussion

The most important finding of our research is the role of anti-dissociative and anti-suicidal functions of NSSI, which have been shown to mediate the association between the lifetime number of NSSIs and lifetime history of suicide attempts. Importantly, mediation appeared to be significant after controlling for the effects of concomitant psychopathology.

Although the relationship of post-traumatic stress disorder and dissociative disorders with NSSIs has been documented, and the association between these disorders and suicidality has been replicated [33], the role of anti-dissociative function has also been highlighted by previous studies. The authors of the meta-analysis investigating the association of dissociative disorders with NSSIs and suicide attempt hypothesised the existence of the “dissociative subtype” among individuals who present both behaviours. The hypothesis was established on the basis of the results, which have shown a higher level of dissociative symptoms in patients reporting suicide attempt and NSSIs [34]. Our results suggest that the presence of the anti-dissociative function might be a risk factor for persistence of NSSIs and might act as an important phenomenon in the association between NSSIs and suicide attempt. Our finding is in line with previous research in young adults [16] and suggest that the anti-dissociative function is a prognostic factor for suicide attempts.

Based on our results, the similar conclusion can be established regarding the anti-suicide function. Though it has been suggested by some authors that the anti-suicide function impacts the association between NSSIs and suicide attempts, our results are the first obtained from a homogenous sample of patients diagnosed with a specific mental disorder. In the most similar clinical sample, 56 inpatients female adolescents were examined [18]. Contrary to our results, the predictive effect of the anti-suicide function on suicide attempts was not found in this sample of patients. However, our sample differs significantly in terms of a history of suicide attempt, where 48.05% of our patients vs. 70% of female inpatients from the study by Kraus reported at least one suicide attempt. The authors discussed high co-occurrence of suicidality and other limitations that might explain non-significant results, including small sample, high comorbidity and co-occurrence of suicidality [18].

## 5. Conclusions

The results of our research indicate the role of NSSI’s functions, however, our findings should also be interpreted with caution due to potential recall bias with respect to reporting the number of NSSIs by participants. Moreover, our study sample was also not large and the sample size and cross-sectional design should be highlighted as main limitations of our study but indeed, the specificity of mental disorders seems to be important and might be responsible for a lack of consistency in the results within different clinical samples, and further might lead to the conclusion that symptomatic differences between mental disorders may affect the effect of various mediators including the impact of specific functions of NSSIs. Both functions (anti-suicide function and anti-dissociative function) represent intrapersonal functions of NSSIs. However, other intrapersonal functions did not impact the relationship between NSSIs and suicide attempt in our sample. Therefore, this fact underlies the role of the anti-suicide function and the anti-dissociative function, and suggest that it is worth focusing on the role of these specific functions as predictors of suicidality in future studies. However, these functions are not listed in the DSM-5 criteria, which may result in insufficient research interest in the future. On the other hand, it seems obvious that the question of whether it is worth differentiating the subtypes of NSSI or determining the severity of NSSI on the risk of suicide remains open. Moreover, in light of findings on the role of anti-suicide function, further research and discussion are needed to clarify the nosological position of the self-injurious behaviours with anti-suicidal function. Indeed, our findings suggest that those behaviours might be strongly related to suicide attempt, placing them in the suicidal behaviours category rather than in non-suicidal behaviours. It is also important to consider the existence of subtypes of self-injuries depending on their association with suicidal behaviours. In order to improve predictive accuracy, further research (prospective, based on larger groups of patients with different mental disorders) is necessary to determine whether the suicidal subtype of self-injuries exists.

## Figures and Tables

**Figure 1 jcm-10-04602-f001:**
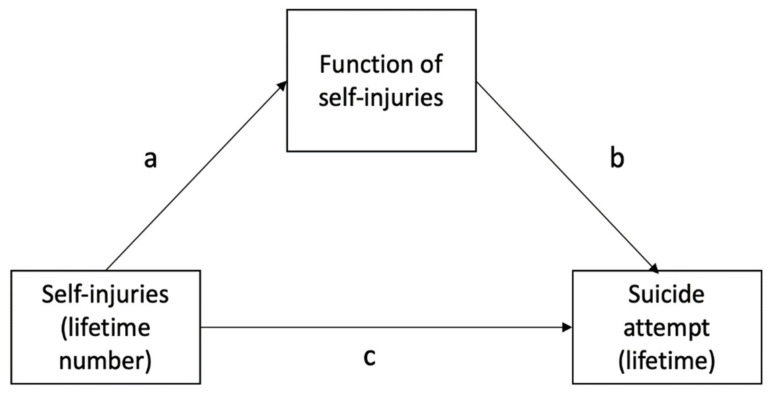
The mediation model tested in the present study. Direct effects are shown: a – direct effect of self-injuries (lifetime number) on function of self-injuries; b – direct effect of function of self-injuries on suicide attempt risk (lifetime) and c – direct effect of self-injuries (lifetime number) on suicide attempt risk (lifetime).

**Table 1 jcm-10-04602-t001:** General characteristics of the sample.

	Suicide Attempt (+), *n* = 37	Suicide Attempt (–), *n* = 40	Statistics
Age, years	14.7 ± 1.1	14.6 ± 1.1	*U* = 789.5, *p* = 0.601
Sex, F/M (%)	31/6	26/14	*χ^2^* = 3.527, *p* = 0.060
Age of self-harm onset, years	10.4 ± 4.5	10.9 ± 4.3	*U* = 568.5, *p* = 0.469
Recent year number of self-injuries	37.6 ± 67.0	33.9 ± 87.9	***U* = 768.5, *p* = 0.048**
Lifetime number of self-injuries	214.0 ± 408.4	151.4 ± 320.6	*U* = 681.5, *p* = 0.702
CDI2—depression	24.6 ± 14.7	17.7 ± 11.5	*t* = 1.385, *p* = 0.178
STAI—trait anxiety	51.0 ± 7.9	43.5 ± 16.1	***t* = 2.521, *p* = 0.014**
STAI—state anxiety	55.1 ± 9.9	46.0 ± 13.7	***t* = 3.237, *p* = 0.002**
SES—self-esteem	21.2 ± 6.2	26.3 ± 5.6	***t* = −3.729, *p* < 0.001**
BPAQ—physical aggression	19.3 ± 7.3	19.7 ± 7.0	*t* = −0.215, *p* = 0.830
BPAQ—verbal aggression	13.5 ± 5.9	14.1 ± 4.8	*t* = −0.495, *p* = 0.622
BPAQ—anger	18.6 ± 6.0	18.3 ± 6.7	*t* = 0.201, *p* = 0.842
BPAQ—hostility	18.7 ± 8.7	19.4 ± 7.7	*t* = −0.387, *p* = 0.700
IVE—adventuresomeness	8.5 ± 3.4	9.2 ± 3.4	*U* = 503.5, *p* = 0.269
IVE — empathy	12.3 ± 3.0	12.4 ± 3.7	*t* = −0.155, *p* = 0.878
IVE—impulsivity	10.3 ± 4.7	11.2 ± 3.4	*t* = −0.977, *p* = 0.332
Affect regulation	3.5 ± 1.7	2.8 ± 1.9	*t* = 1.658, *p* = 0.102
Interpersonal boundaries	1.7 ± 1.5	1.6 ± 1.6	*U* = 564.5, *p* = 0.612
Self-punishment	3.3 ± 2.2	2.3 ± 2.0	*U* = 670.0, *p* = 0.056
Self-care	2.0 ± 1.6	1.5 ± 1.7	*U* = 696.5, *p* = 0.128
Anti-dissociation feeling generation	2.7 ± 1.9	1.7 ± 1.9	***U* = 750.0, *p* = 0.030**
Anti-suicide	2.9 ± 1.7	1.7 ± 1.6	***U* = 794.0, *p* = 0.007**
Sensation seeking	1.2 ± 1.1	1.5 ± 1.5	*U* = 540.0, *p* = 0.646
Peer bonding	1.0 ± 1.4	1.7 ± 1.9	*U* = 479.5, *p* = 0.203
Interpersonal influence	1.8 ± 1.7	1.6 ± 1.6	*U* = 0.620, *p* = 0.578
Toughness	2.5 ± 1.7	1.7 ± 1.8	*U* = 721.0, *p* = 0.070
Marking distress	1.2 ± 1.6	1.1 ± 1.3	*U* = 584.0, *p* = 0.914
Revenge	1.4 ± 1.6	1.6 ± 1.7	*U* = 541.0, *p* = 0.656
Autonomy	1.6 ± 1.6	1.5 ± 1.7	*U* = 597.0, *p* = 0.790

Data expressed as mean ± SD or *n* (%). Significant differences (*p* < 0.05) were marked with bold characters. Abbreviations: BPAQ—the Buss–Perry Aggression Questionnaire; CDI2—the Children’s Depression Inventory 2; EI—emotional intelligence; IVE—the Eysenck’s Impulsivity Inventory; PEIQ—the Popular Emotional Intelligence Questionnaire; QADS—the Questionnaire for the Assessment of Disgust Sensitivity; SES—the Rosenberg Self-Esteem Scale; STAI—the State-Trait Anxiety Inventory.

**Table 2 jcm-10-04602-t002:** Correlations between lifetime number of self-injuries and their function.

Affect regulation	***r* = 0.551, *p* < 0.001**
Interpersonal boundaries	*r* = 0.129, *p* = 0.301
Self-punishment	***r* = 0.501, *p* < 0.001**
Self-care	***r* = 0.495, *p* < 0.001**
Anti-dissociation feeling generation	***r* = 0.474, *p* < 0.001**
Anti-suicide	***r* = 0.495, *p* < 0.001**
Sensation seeking	*r* = 0.042, *p* = 0.732
Peer bonding	*r* = −0.082, *p* = 0.501
Interpersonal influence	*r* = 0.154, *p* = 0.206
Toughness	***r* = 0.418, *p* < 0.001**
Marking distress	*r* = 0.030, *p* = 0.809
Revenge	*r* = 0.195, *p* = 0.108
Autonomy	*r* = 0.210, *p* = 0.083

Significant correlations were marked with bold characters (*p* < 0.05).

**Table 3 jcm-10-04602-t003:** Results of mediation analysis.

Mediator	Effect	Statistics
Anti-dissociation feeling generation	Effect of lifetime number of NSSIs on mediator (a)	**B = 0.0012, SE = 0.006, 95%CI = 0.0001–0.0024**
	Effect of mediator on lifetime history of suicide attempts (b)	B = 0.2591, SE = 0.1383, 95%CI = −0.0119–0.5300
	Direct effect of lifetime number of NSSIs on lifetime history of suicide attempts (c)	B = 0.002, SE = 0.007, 95%CI = −0.0012–0.0015
	Indirect effect of lifetime number of NSSIs on lifetime history of suicide attempts (through mediator)	**B = 0.0003, SE = 0.0005, 95%CI = 0.001–0.0018**
Anti-suicide	Effect of lifetime number of NSSIs on mediator (a)	**B = 0.0056, SE = 0.0027, 95%CI = 0.0002–0.0110**
	Effect of mediator on lifetime history of suicide attempts (b)	**B = 0.425, SE = 0.1646, 95%CI = 0.1024–0.7477**
	Direct effect of lifetime number of NSSIs on lifetime history of suicide attempts (c)	B = 0.0024, SE = 0.0034, 95%CI = −0.0083–0.0048
	Indirect effect of lifetime number of NSSIs on lifetime history of suicide attempts (through mediator)	**B = 0.0023, SE = 0.0028, 95%CI = 0.002–0.0108**

Significant effects (95%CI does not include zero) were marked with bold characters. a, b and c refer to paths shown in Figure 1.

**Table 4 jcm-10-04602-t004:** Results of mediation analysis after excluding participants with lifetime number of NSSIs higher than 100.

Mediator	Effect	Statistics
Anti-dissociation feeling generation	Effect of lifetime number of NSSIs on mediator (a)	**B = 0.0266, SE = 0.0091, 95%CI = 0.0081–0.0450**
	Effect of mediator on lifetime history of suicide attempts (b)	**B = 0.4872, SE = 0.2115, 95%CI = 0.0726–0.9018**
	Direct effect of lifetime number of NSSIs on lifetime history of suicide attempts (c)	B = −0.0133, SE = 0.0138, 95%CI = −0.0403–0.0137
	Indirect effect of lifetime number of NSSIs on lifetime history of suicide attempts (through mediator)	**B = 0.0129, SE = 0.0105, 95%CI = 0.0024–0.0435**
Anti-suicide	Effect of lifetime number of NSSIs on mediator (a)	**B = 0.0249, SE = 0.0085, 95%CI = 0.0078–0.0421**
	Effect of mediator on lifetime history of suicide attempts (b)	**B = 0.6163, SE = 0.2423, 95%CI = 0.1415–1.0911**
	Direct effect of lifetime number of NSSIs on lifetime history of suicide attempts (c)	B = −0.0166, SE = 0.0148, 95%CI = −0.0456–0.0124
	Indirect effect of lifetime number of NSSIs on lifetime history of suicide attempts (through mediator)	**B = 0.0154, SE = 0.0114, 95%CI = 0.0030–0.0475**

Significant effects (95%CI does not include zero) were marked with bold characters. a, b and c refer to paths shown in Figure 1.

## Data Availability

The data presented in this study are available on request from the corresponding author.

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
