# Peer review of "Self-Injuries and Their Functions with Respect to Suicide Risk in Adolescents with Conduct Disorder: Findings from a Path Analysis"

_jcm, 2021, doi:10.3390/jcm10194602_

Round 1

Reviewer 1 Report

Please see comments in attached pdf file.

Author Response

We would like to thank you for your valuable comments.

We have corrected the manuscript according to your suggestion.

Reviewer 2 Report

This study showed that anti-dissociation and anti-suicide function mediate the association between NSSIs and suicide attempts in life in adolescents with conduct disorder. The association between NSSIs and suicide attempts is an essential topic in adolescent health epidemiology. Significantly, this study examined the mediation effects in detail in a specific disease group (i.e., conduct disorder).
Although the study seems to be of a standard published in this journal, some concerns require revision. I recommend that the following issues be re-examined and revisited.

Major concerns

METHOD
The definition of NSSIs in this study should be included. Specifically, what behaviors were considered NSSIs?
In addition, the validity of obtaining information on NSSIs through self-reports by children is questionable. In particular, if objective indicators, such as the number of times, are obtained retrospectively through self-reports, there is a high possibility that bias may have been introduced. There are concerns about whether "several hundred times" is an accurate number.

RESULT
The text should state that there were 77 NSSIs (only stated in the abstract).
It should also be stated whether there was a difference in basic demographics between the 77 with NSSIs and the 138 without NSSIs.

In the mediation analysis, it would be helpful to conduct a sensitivity analysis that excludes extreme cases of NSSIs. The reason is that the number of NSSIs in Table 1 shows an apparent floor effect, suggesting an extreme outlier in the number NSSIs, and that, as mentioned above, the reliability and validity of the number of "hundreds of times" is questionable.

DISCUSSION
Assuming that anti-dissociation and anti-suicide mediate NSSIs and suicide attempts, we should also discuss the effect size, i.e., how much the risk increases quantitatively. According to the results of this study (anti-dissociation: B = 0.0003, anti-suicide: B = 0.0023), even if NSSIs increase 200 times, the increase in suicide attempts is 0.06 and 0.46. Can we say that this is a realistic predictor of suicide attempts, especially for anti-dissociation?

Minor concerns

ABSTRACT
There seems to be insufficient information on why we focused on CD in this study.

METHOD
Line 224
"Figure 1" does not seem to be present in this draft. 

DISCUSSION
Line 274
Is "lifetime number of suicide attempts" a misspelling of "a lifetime history of suicide attempts"?

Author Response

This study showed that anti-dissociation and anti-suicide function mediate the association between NSSIs and suicide attempts in life in adolescents with conduct disorder. The association between NSSIs and suicide attempts is an essential topic in adolescent health epidemiology. Significantly, this study examined the mediation effects in detail in a specific disease group (i.e., conduct disorder).
Although the study seems to be of a standard published in this journal, some concerns require revision. I recommend that the following issues be re-examined and revisited.

Thank you for your valuable comments. Below point-by-point responses are provided.

Major concerns

METHOD
The definition of NSSIs in this study should be included. Specifically, what behaviors were considered NSSIs?

Response: Self-injuries were classified as NSSIs if they met the following characteristics: 1. Deliberate self-injury of body tissues without suicidal intent during the act; 2. Self-injuries with at least one of the methods: cutting, burning, scratching, banging, excessive rubbing and hitting. 3. At least one of the following functions of self-injuries was indicated by the patient: relieving an interpersonal difficulty, reducing negative emotions and/or inducing positive feelings.

Therefore, the following answers were required in the semi-structured interview about NSSIs:

  • “Yes” to question 1 and “No” to question number 2;
  • At least one of the following: cutting, burning, picking, hitting and/or excessive rubbing indicated as the answer to question number 3.
  • One of the following answers to question number 5 (in subsection statements): to alleviate anxiety/sadness/tension, to manage bad feelings, to punish or control somebody, to get someone’s attention, to punish yourself and/or to feel better.

In addition, the validity of obtaining information on NSSIs through self-reports by children is questionable. In particular, if objective indicators, such as the number of times, are obtained retrospectively through self-reports, there is a high possibility that bias may have been introduced. There are concerns about whether "several hundred times" is an accurate number.

Response: All data regarding NSSIs were collected using the semi-structured interview (provided in the manuscript). The interview was conducted by trained psychologist or psychiatrist to assess the frequency of NSSIs (including the number of NSSIs in different time periods) and the functions of NSSIs, as precisely as possible.   

But indeed , the recall bias cannot be excluded therefore we added in Discussion “Our findings should be interpreted with caution due to potential recall bias with respect to reporting the number of NSSIs by participants”.  

RESULT
The text should state that there were 77 NSSIs (only stated in the abstract).

Response: The number of adolescents who engaged in NSSIs was added in the results section.

It should also be stated whether there was a difference in basic demographics between the 77 with NSSIs and the 138 without NSSIs.

Response: Differences in basic demographics (age and sex) were provided in the results.

In the mediation analysis, it would be helpful to conduct a sensitivity analysis that excludes extreme cases of NSSIs. The reason is that the number of NSSIs in Table 1 shows an apparent floor effect, suggesting an extreme outlier in the number NSSIs, and that, as mentioned above, the reliability and validity of the number of "hundreds of times" is questionable.

Response: Thank you for rising this point. We have performed sensitivity analysis that excluded participants reporting lifetime number of NSSIs greater than 100. We obtained similar findings to those in the pooled analysis (see Table A2 for details).

DISCUSSION
Assuming that anti-dissociation and anti-suicide mediate NSSIs and suicide attempts, we should also discuss the effect size, i.e., how much the risk increases quantitatively. According to the results of this study (anti-dissociation: B = 0.0003, anti-suicide: B = 0.0023), even if NSSIs increase 200 times, the increase in suicide attempts is 0.06 and 0.46. Can we say that this is a realistic predictor of suicide attempts, especially for anti-dissociation?

Response: It should be noted that B in mediation analyses cannot be interpreted as effect size estimates. Importantly, there is also no consensus as to which measures of effect size should reported in mediation analyses (Wen and Fan, Psychological Methods 2015;20:193-203; Preacher and Kelley, Psychological Methods 2011;16:93-115).

Minor concerns

ABSTRACT
There seems to be insufficient information on why we focused on CD in this study.

Due to limited words allowed in Abstract short explanation “which shares some common characteristics with NSSIs” was added, and common  characteristics were mentioned in the Introduction.

“Conduct disorder (CD) seems to be one of the psychiatric disorders that shares some common characteristics with NSSIs including the following: aggression (which was  suggested as one of the crucial risk factors of NSSI),  a typical age of onset in adolescence, interpersonal difficulties as well as the association with a history of childhood maltreatment  an invalidating family environment  and  low socioeconomic status (Fliege 2009, Kaess et al., 2013, Mars et al., 2014, Murray et  Farrington 2010).”

METHOD
Line 224
"Figure 1" does not seem to be present in this draft. 

Figure 1 has been added.

DISCUSSION
Line 274
Is "lifetime number of suicide attempts" a misspelling of "a lifetime history of suicide attempts"?

Indeed, “lifetime number of suicide attempts” was a misspelling.

Round 2

Reviewer 2 Report

In the revised manuscript, all comments from the reviewers were appropriately revised.

In particular, the rationale for why the author is focusing on the relationship between NSSI and conduct disorder has been made more explicit. So, It made it easier for the reader to understand the value of this study.